# Genome-Wide Signal Selection Analysis Revealing Genes Potentially Related to Sheep-Milk-Production Traits

**DOI:** 10.3390/ani13101654

**Published:** 2023-05-16

**Authors:** Ruonan Li, Yuhetian Zhao, Benmeng Liang, Yabin Pu, Lin Jiang, Yuehui Ma

**Affiliations:** 1State Key Laboratory of Animal Biotech Breeding, Institute of Animal Sciences, Chinese Academy of Agricultural Sciences (CAAS), Beijing 100193, China; liruonan1998424@163.com (R.L.); jianglin@caas.cn (L.J.); 2Key Laboratory of Animal Genetics Breeding and Reproduction, Ministry of Agriculture and Rural Affairs, Chinese Academy of Agricultural Sciences (CAAS), Beijing 100193, China; 3Teaching and Research Centre (TERRA), Gembloux Agro-Bio Tech, University of Liège, 5030 Gembloux, Belgium

**Keywords:** dairy sheep, whole-genome resequencing, RT-qPCR, milk production, gene expression

## Abstract

**Simple Summary:**

In our research, candidate genes related to sheep-milk production were revealed by a genome-resequencing analysis and a genome-signal-selection analysis, and a RT-qPCR experiment was performed to prove the expression levels of these candidate genes, the results showed that the FCGR3A gene’s expression level had a significant negative relationship with sheep-milk production.

**Abstract:**

Natural selection and domestication have shaped modern sheep populations into a vast range of phenotypically diverse breeds. Among these breeds, dairy sheep have a smaller population than meat sheep and wool sheep, and less research is performed on them, but the lactation mechanism in dairy sheep is critically important for improving animal-production methods. In this study, whole-genome sequences were generated from 10 sheep breeds, including 57 high-milk-yield sheep and 44 low-milk-yield sheep, to investigate the genetic signatures of milk production in dairy sheep, and 59,864,820 valid SNPs (Single Nucleotide Polymorphisms) were kept after quality control to perform population-genetic-structure analyses, gene-detection analyses, and gene-function-validation analyses. For the population-genetic-structure analyses, we carried out PCA (Principal Component Analysis), as well as neighbor-joining tree and structure analyses to classify different sheep populations. The sheep used in our study were well distributed in ten groups, with the high-milk-yield-group populations close to each other and the low-milk-yield-group populations showing similar classifications. To perform an exact signal-selection analysis, we used three different methods to find SNPs to perform gene-annotation analyses within the 995 common regions derived from the fixation index (F_ST_), nucleotide diversity (Ɵπ), and heterozygosity rate (ZHp) results. In total, we found 553 genes that were located in these regions. These genes mainly participate in the protein-binding pathway and the nucleoplasm-interaction pathway, as revealed by the GO- and KEGG-function-enrichment analyses. After the gene selection and function analyses, we found that *FCGR3A*, *CTSK*, *CTSS*, *ARNT*, *GHR*, *SLC29A4*, *ROR1*, and *TNRC18* were potentially related to sheep-milk-production traits. We chose the strongly selected genes, *FCGR3A*, *CTSK*, *CTSS*, and *ARNT* during the signal-selection analysis to perform a RT-qPCR (Reale time Quantitative Polymerase Chain Reaction) experiment to validate their expression-level relationship with milk production, and the results showed that *FCGR3A* has a significant negative relationship with sheep-milk production, while other three genes did not show any positive or negative relations. In this study, it was discovered and proven that the candidate gene *FCGR3A* potentially contributes to the milk production of dairy sheep and a basis was laid for the further study of the genetic mechanism underlying the strong milk-production traits of sheep.

## 1. Introduction

The sheep is one of the earliest domesticated-farm-animal species and has experienced evolution and domestication over thousands of years [1]. Dairy sheep are traditionally farmed in southern Europe (France, Italy, Spain, Greece), central Europe (Hungary and the Czech and Slovak Republics), eastern Europe (Romania and Ukraine), and countries in the Middle East, such as Turkey and Iran [2]. Organized sheep-breeding programs were developed from at least the 1960s [3]. The most efficient selection scheme for local dairy sheep is based on the pyramidal management of the population, with the breeders of the nucleus flocks at the top, and pedigree, official milk recording, AI (artificial insemination) breeding, controlled natural mating, and breeding-value estimation (i.e., BLUP) are carried out to make genetic progress. In 2013, dairy small ruminants accounted for a minor part of the total agricultural output in France, Italy, and Spain (0.9 to 1.8%) and a larger part in Greece (8.8%) [4]. In these European countries, the dairy-sheep industry is based on local breeds and crossbreeds raised under semi-intensive and intensive systems, and it is concentrated in a few regions. While with the development of dairy-sheep breeding and the emergence of dairy products emerging, dairy ruminants have become a major part of European agricultural income [5,6]. The average flock size varies from small to medium (140 to 333 ewes/farm), and the average milk yield ranges from low to middle (170 to 500 L/ewe) [7], showing substantial space for improvement in relation to cows’ milk [8]. Furthermore, sheep milk has higher protein, fat, lactose, total non-fat solids, and ash contents and a higher nutritional value than cows’ and goats’ milk [9], which makes it suitable for processing into various types of dairy products. Most sheep milk is sold to industries and then processed into traditional cheese products, most of which are made into Protected Denomination of Origin (PDO) cheeses for gourmets. The animals’ udder health is also critical in sheep-milk quantity and quality. Mastitis is an inflammation of the mammary gland that is usually caused by pathogens, mainly bacteria, which develop in the udder tissue after infections in the teat canal. It is one of the most prevalent and costly diseases in the dairy industry due to the significant reductions in milk production and physical harm that it causes [10,11,12]. In previous studies, some quantitative trait loci (QTLs) and SNPs associated with SCC (somatic cell counts) that serve as markers of mastitis were identified based on linkage-disequilibrium analyses of different dairy-sheep breeds [10,13,14], leading to significant improvements in ewes’ ability to resist mastitis [15,16].

The dramatic decreases in the costs of whole-genome sequencing (WGS) and RT-qPCR experiments on animals have made it possible to scan the complete genomes of thousands of animals. Using genome information makes it possible to explain parts of the total genetic variance that are difficult to measure, such as low-heritability, sex-limited, and postmortem traits. To help clarify the link between the genome sequence and real data of important traits, this research was designed to gain a better understanding of sheep lactating genes by comparing the SNP-variant information on different sheep breeds with significantly different dairy-production characteristics. These breeds significantly differ at two dairy levels, since there are high-milk-yield (East Friesian sheep, 700 kg/lactation, Dairy Meade sheep, 500 kg/lactation and Awassi sheep) and low-milk-yield sheep breeds (Hu sheep, small-tailed Han sheep, and Churra sheep) [17]. To this end, we performed a whole-genome sequencing (WGS) analysis of these sheep breeds, as we found that in previous studies, little research was performed using whole-genome sequencing on these dairy sheep populations. First, we re-sequenced the crossed breed of Dairy Meade sheep and small-tailed Han sheep. The information on this sheep breed will be uploaded to a public genome-sequence database for researchers to use freely (NCBI). In our research, a genetic-structure analysis was performed to establish the relationships and geographical distances between 10 sheep breeds, and then genome-wide-signal scans were carried out to identify significant genes associated with sheep-milk production. The genes validated by the RT-qPCR results and their relationship with milk production represent significant progress in our knowledge of the gene-expression levels of lactating sheep and, thus, provide potentially valuable information for future dairy-sheep studies.

## 2. Material and Methods

### 2.1. Animals

One hundred and one sheep samples were sequenced in this study. This step was approved by the Animal Ethics Committee of China. In total, 41 sheep-ear samples were provided by the M-Natural Animal Husbandry Technology Company, including 36 sheep with high levels of milk production and 5 sheep with low levels of milk production. The high-milk-yield populations included Dairy Meade (DM) sheep (11), the crossbreed of Dairy Meade sheep and small-tailed Han sheep DM (F1) (15), and the crossbreed of DM (F1) and small-tailed Han sheep, DM (F2) (10). The low-milk-yield population consisted of small-tailed Han sheep (STHS) (5). The sequence data of the 60 remaining sheep were downloaded from (https://www.ncbi.nlm.nih.gov/sra, accessed on 15 March 2022). These sheep were from three high-milk-yield breeds, East Friesian (EFR) (10), Awassi (AWS) (2), and Dairy Meade (9), and four low-milk-yield breeds: Hu (HS) (14), Fin (FS) (9), Suffolk (SFK) (10), and Churra (CS) (6) (Table 1).

### 2.2. Whole-Genome-Resequencing Analysis

We sequenced 41 sheep genomes at an average depth-of-coverage of 10X, which contained 4 sheep breeds. To this end, genomic DNA was extracted from ear tissue using the Wizard^®^ Genomic DNA purification kit (Promega, A1125, Madison, WI, USA). The A260/A280 ratio using Nano-Drop ND-2000 (Thermo Fisher Scientific, MA, USA) was used to check the quality and integrity of the extracted DNA and agarose-gel electrophoresis was used to check its quality. Good-quality DNA from each collected sample was sequenced on the MGI-SEQ2000 platform from the Beijing Compass Biotechnology Company (Beijing, China). The satisfactory sequence data obtained were used to characterize individual genomes at a minimum of 10X depth of coverage, with more than 30 G of raw data from each sample. After quality control, data were used to perform genome mapping and SNP calling. Resequencing data from a further 60 sheep, downloaded from NCBI, were analyzed by Plink parameters on Linux (San Francisco, CA, USA) (Version V1.90). The details are listed in Section 2.3. All high-quality double-trimmed read pairs were aligned against the reference assembly Oar v.4.0 genome [29] (https://www.ncbi.nlm.nih.gov/assembly/GCA_000298735.2, accessed on 15 March 2022) using BWA software (https://sourceforge.net/projects/bio-bwa/files/, accessed on 15 March 2022) (version: 0.7.12). Paired-end reads that mapped to exactly the same position on the reference genome were deleted by the Mark Duplicates in Picard (picard-tools-1.56, at http://picard.sourceforge.net, accessed on 15 March 2022). Additional realignment of indels and SNPs was performed by using the Genome Analysis Toolkit (GATKV4.0) (https://gatk.broadinstitute.org/hc/en-us, accessed on 15 March 2022) [30] and sequence alignment (SAM tools) [31]. The GATK was used to identify SNP variation in each individual sample. The SNPs that did not meet the following criteria were excluded: (1) SNP call rate > 99.6%; (2) minor-allele frequency > 0.01; (3) missing rate is lower than 0.1; and (4) all loci followed the Hardy–Weinberg rule; (5) linkage-disequilibrium loci were excluded (R^2^ < 2). All SNPs were annotated using the Bio Mart 4.0 (https://asia.ensembl.org/info/data/biomart/index.html, accessed on 15 March 2022) [32] based on the gene-reference genome provided by the Oar v.4.0 genome from NCBI.

### 2.3. Population Structures and Phylogenetic Analysis

Population structures among all sheep breeds were investigated using a total of 59,864,820 high-quality SNPs. The VCF (Visual Component Framework) files were transformed in the PLINK format by using VCF Tools (https://vcftools.sourceforge.net/, accessed on 15 March 2022) [33], and the SNP filtering was undertaking under the following conditions: minor-allele frequency (MAF) greater than 0.05, call-out rate higher than 0.9, missing-genotype rate > 0.05, and Hardy–Weinberg rules > 1 × 10^6^. Pruning was carried out within windows of 50 SNPs and in 5 steps, using the indep-pairwise 50 5 2 parameters in PLINK [34] (plink --file qc --indep 50 5 2 --chr-set 27 --recode --out qc_prune) to obtain linkage-disequilibrium SNPs. This provided a total of 4,751,642 unlinked SNPs for PCA, ADMIXTURE, and neighbor-joining-tree-construction analysis. The PCA was conducted using the GCTA software (https://yanglab.westlake.edu.cn/software/gcta/, accessed on 15 March 2022) [35]; the first two components were plotted using with the R program gg plot package (https://www.r-project.org/, accessed on 15 April 2022) (version 4.1.0). Individual ancestry origins were predicted using replicates in the ADMIXTURE (http://software.genetics.ucla.edu/admixture/download.html, accessed on 15 April 2022) (version 1.3.0) [36], and assuming 2 to 4 ancestral populations (K), with the smallest CV error as the correct result. Neighbor-joining tree was constructed using the PHYLIP v3.69 (https://evolution.genetics.washington.edu/phylip.html, accessed on 15 April 2022) [37] based on the pairwise genetic distance matrix revealed by the PLINKv1.9 (plink1 --sheep --file prune --cluster --distance-matrix –out), and then MEGA7 (https://www.megasoftware.net/, accessed on 15 April 2022) [38] was used to construct the phylogenetic trees.

### 2.4. Genome-Wide Signal-Selection Scan and Gene Annotation

We used all SNPs that passed quality control to detect signatures of selection in high- and low-milk-yield sheep within 101 sheep genomes. Genome-wide signal-selection analysis was performed by using F_ST_ fixation indices (population-differentiation value), (θπ), the nucleotide-diversity ratio (θπ-HMY/θπ-LMY), HMY (high milk yield), LMY (low milk yield), and the transformed heterozygosity score (ZH_P_). The window-based ZH_P_ method was calculated by the formula ZH_P_ = (Hp − *μ*Hp)/*σ*Hp, where *μ* is the overall average signal value of heterozygosity and *σ* is the standard deviation of all windows of each group. High-milk-yield group included 57 samples from five breeds (Dairy Meade sheep = 20, DM (F1) = 15, DM (F2) = 10, East Friesian sheep = 10, and Awassi sheep = 2), while low-milk-yield group included 44 individuals across five breeds (STHS = 5, Hu sheep = 14, Fin sheep = 9, Churra sheep = 6 and Suffolk sheep = 10). The F_ST_ fixation, ZHp [39], and log_2_θπ (high/low), were calculated within 100-kb sliding windows and 10-kb steps to obtain overlapping regions. The variation regions within the highest 5% of all three statistics were considered as candidate selections, and then candidate genes were annotated by a genomic-database search and annotating engine, Bio Mart (https://asia.ensembl.org/info/data/biomart/index.html, accessed on 15 March 2022) [40].

### 2.5. Gene Ontology and Kyoto Encyclopedia of Genes and Genomes

Functional enrichments for Gene Ontology (GO) terms and Kyoto Encyclopedia of Genes and Genomes (KEGG) pathway analyses were performed using g: Profiler (https://biit.cs.ut.ee/gprofiler/gost, accessed on 1 June 2022) for all selected genes. The enrichment significance was assessed using a condition for g: SCS ‘Set Counts and Sizes’, and the *p* value was set at <0.05.

### 2.6. Validation of RT-qPCR Experiment

We randomly selected 11 Lacaune sheep in from a farm in Belgium (https://lesfauveslaineux.wordpress.com/le-troupeau/, accessed on 1 September 2022) to determine their genes’ expression levels. These were frequently selected in our research, since Lacaune sheep are among the most famous and heavily produced dairy sheep in France, and imported into Belgium. This farm efficiently recorded the milk-production data from different milk lactations and information on newly lambing ewes. Fresh milk was sampled from 11 newly lambed ewes. From each ewe, 50 mL of milk was sampled in non-RNA se tubs. All these steps complied with the requirements of European Animal Welfare Committee. The milk was stored at 4 °C during transport from farm to laboratory and processed immediately for somatic cell isolation. Subsequently, 50 uL of 0.5 M EDTA was added for each 50-mL milk sample. The samples were then centrifuged for 10 min at 2000× *g*, after which the supernatant containing cream and skin milk was removed, followed by washing in 10 mL PBS. Samples were then centrifuged a second time, the maximum amount of supernatant was removed, and the lysed samples were stored at −80 °C. The RNA was extracted according to the procedure of the isolation of RNA from non-fibrous tissue (Promega Inc., Madison, WI, USA), after which concentration and A260/A280 of RNA were tested; the standards were concentration >10 ng/ul and 2.2 > A260/A280 > 1.8. Satisfactory RNA was used to perform reverse transcription in accordance with Protocole reverse transcription (Promega Inc., Madison, WI, USA). The conditions of reverse transcription were as follows: 25 °C, 5 min; 42 °C, 60 min; 70 °C, 15 min, 4 °C, 20 min; 4 cycles in total. The cDNA was stored at −20 °C. Quantitative real-time PCR was performed using DNA, premiers were designed to amplify target genes and reference genes, reference genes were selected from NCBI database, which were similarly expressed in all tissues, the genes’ premier pairs for RT-qPCR were designed by Eurogentec CO., (Brussel, Belgium), and RT-qPCR analyses were performed using Rotor Gene 6000 system (Corbett) and SYBR green (Gembloux, Belgium). Each amplification reaction contained 2 μL of cDNA and 18 μL of SYBR master mix (Thermo Fisher Scientific, MA, USA), including 500 nM of primers at a final volume of 20 μL. Reactions were performed as follows: 95 °C for 5 s, 60 °C for 30 s, 72 °C for 45 s, for 40 cycles in total, with data collection at the annealing step.

### 2.7. The Relationship between Gene Expression and Milk Production

Candidate genes from RT-qPCR experiment were used to perform a linear correlation analysis between genes’ relative expression values and milk production. The milk-production recordings were from Belgian farm with the same Lacaune sheep as those in 2022 (Appendix A), as well as the production data from 2015 to 2021 (Appendix A). We used the BLUPF90+ (http://nce.ads.uga.edu/html/projects/programs/, accessed on 15 March 2023) program to obtain estimated values (similar to estimated breeding values) of milk-production traits of each ewe from 2015 to 2021. Usually, higher estimated values meant the animal had stronger production traits, so we used these values to conduct a linear analysis with ΔCT gene values and actual milk production from 2022, as well as the estimated values from period of 2015–2021 to show their relationships. Animal model is needed in BLUPf90+. In our research, the animal model was as follows: y_ijk_ = A_i_ + S_j_ + βx_ijk_ + e_ijk_, where y_ijk_ is observed milk production, A_i_ is fixed effect of lactation days, lactation number, birth year, and milking times, S_j_ is random effect of animals, βx_ijk_ is regression coefficient of variance of 1.0, and e_ijk_ is residual effect. The estimations were given after data analysis using BLUPF90+ program.

## 3. Results

### 3.1. Genomic Variants and Principal Component Analysis

We combined the genome sequences of 101 high- and low-production dairy sheep. The genomes were mapped and the SNPs were called using PLINK. After the joint calling and quality control, we detected 4,751,642 SNPs in total. To study the population stratification, three different approaches were employed, of which PCA is the first to be discussed here. The intention behind the use of PCA was to reduce the dimensionality of the genomic relationship matrix so that individuals (and breeds) were separated along different principal components (PCs). The first three principal components were able to explain most of the variation: PC1 explained about 50.26%, PC2 explained about 23.51%, and PC3 explained about 14.95%. Principal component 1 (PC1) differentiated the breeds in this study into three main clusters (Figure 1A): Cluster1 (high-yield breeds, EFR, DM, DMF1, and DMF2); Cluster2 (Mongolian low-yield breeds, HS, STHS, CS, and AWS) and Cluster3 (low-yield breeds, FS and SFK). However, overlap was observed among the DM sheep, DMF1 and DMF2, within Cluster 1. The reason for this may have been that these sheep have extremely close genetic relations. The PCA separated the different sheep breeds in this study, with SFK the most clearly separated due to its longer distance from and non-blood relations with the other sheep. In the PCA plots, the high-yield breeds were closer to the Mongolian sheep than to the low-yield sheep, and in the high-yield cluster, the EFR sheep were clearly separated from the DM sheep. The separation of the clusters corresponded to the geographical origins and blood mixes of these sheep breeds, and may have also revealed their genetic distance.

### 3.2. Neighbor-Joining-Tree Analysis

In this study, the phylogenetic neighbor-joining tree divided the high- and low-production groups into nine different clusters (Figure 1B). The nine main branches were as follows: DM, DMF1, DMF2, EFR, CS, FS, HS, STHS, and SFK. According to the tree, we observed that the DM and DMF2 sheep branches were close to each other and that some of the DM, DMF1, and DMF2 sheep were mixed together, indicating their genetic similarity and probable sharing of a single ancestral line. The DM sheep is the hybrid offspring of EFR breeds and New Zealand sheep [41], so it was the second-nearest to the EFR sheep. Our phylogenetic results showed two breeds (DMF1 and DMF2) in the different sub-branches, which might have been due to the hybridization between the DM breeds and the STHS breeds. For the HS and STHS sheep, all the Mongolian sheep had similar genetic information, and their branches were close to each other. Similar to the PCA, there was a close genetic distance between the FS and SFK sheep, since there are shorter geographical distances between them. In the NJ-tree picture, the Awassi breed was not clearly separated, possibly due to its small numbers. It should be noted that the results of the phylogenetic analysis were reliable due to the large number of SNPs from the 10 sheep breeds used in this study.

### 3.3. Ancestor Analysis

To further explore the population structure among individuals from the different breeds in this study, a model-based hierarchical clustering analysis was undertaken for K-values of 2–7 (with K the user-defined number of biological ancestral populations). The cross-validation (CV) estimates revealed the K-value of 3 to be the best fit for the nine populations; it demonstrated the lowest CV error among all the K values. The bar plot of the ADMIXTURE results from the K-values of 2 and 3 is presented in Figure 1C. The analysis with K = 2 separated the high-milk-yield sheep breeds from the low-yield breeds. When K = 3, the FS and SFK were separated from the other breeds in the low-yield group. The DM and EFR sheep had the highest milk yield among all the populations, the Mongolian CHS, HS, and STHS sheep had average milk production [26], and the FS and SFK sheep produced the least milk. It is known that FS and SFK sheep are clearly separated from the other two sheep populations, and this was validated by the results obtained from the PCA analysis and the admixture analysis.

### 3.4. Signal-Selection Analysis and Gene Ontology

We scanned the genomes of 101 high- and low-milk-yield sheep for signals of positive selection with different traits. To achieve this, we calculated three complementary statistics along the sheep reference genome Oar v.4.0 (https://www.ncbi.nlm.nih.gov/assembly/GCA_000298735.2, accessed on 15 March 2022) using 100-kb-long sliding windows and 15-kb step sizes. The first statistic was the population-differentiation index F_ST_, for identifying genomic regions with different allelic frequencies between high and low groups. The second statistic measured the differences in nucleotide diversity between the two groups (θπ(high/low)), with high or low θπ values indicating positive selection of different milk-production sheep, respectively. The third measure was also used to calculate the selected candidate SNPS using the index of the transformed heterozygosity score (ZH_P_) (Figure 2). With regard to the limiting of the false-positive identifications, we considered the 5% most heavily overlapping regions from all three scans, which provided a total of 995 overlapping SNP regions and 553 protein-coding genes and small RNA. The selection candidates identified in the high- and low-yield groups are provided in Appendix A. Most of the protein-coding genes were significantly enriched using the functional gene ontology and KEGG categories related to the protein binding, RNA binding, molecular transport, and cytokine-receptor activity (*p* = 9.23 × 10^−3^) (Figure 3, Appendix A). Importantly, through the gene annotation, the three highest 5% genomic regions providing the strongest signatures of selection containing *FCGR3A*, *CTSK*, *CTSS*, *ARNT*, *GHR*, and *SLC29A4* were selected as the strongest annotation genes by comparing the high- and low-milk-yield groups.

### 3.5. RT-qPCR Results

We chose the four genes that were most frequently selected by the signature-selection analysis shown in Figure 2, which were *FCGR3A*, *CTSK*, *CTSS*, and *ARNT*, according to the NCBI database and previous studies. We chose *GAPDH* gene as reference gene to perform RT-qPCR validation. The premier sequence is listed in Table 2. All the gene-expression values were derived from 11 ewes; the CT and ΔCT values of each gene are listed in Appendix A, with ΔCT values meaning the relative expression of each gene.

### 3.6. The Relationship between Gene Expression and Milk Production

The milk-production data were sourced from the Belgian farm. They covered the average daily milk production from four months in 2022: September, October, November, and December (Appendix A). A linear analysis of the ΔCT values of each gene was performed using the daily average milk production from 2022 and values estimated using the program of BLUPf90+ (Appendix A), and the correlations between daily milk production and the estimated values are also shown in Figure 4. The results showed that the *FCGR3A* gene had significant correlations with milk production and estimate values. The higher ΔCT values meant lower expression values during the lactation period. It was revealed that the *FCGR3A* gene had a significantly negative relationship with milk production and the estimate values (R^2^ = 0.8452, R^2^ = 0.7382); furthermore, as predicted, the relationship between actual milk production in 2022 and the estimate value was positive (R^2^ = 0.6988. The *CTSK*, *CTSS*, and *ARNT* did not show significant differences between their expression values, the milk production values, and the estimate values, but the correlation coefficients between actual milk production in 2022 and the estimate values for the 2015–2021 milk production were positive (R^2^ = 0.6988, R^2^ = 0.628, R^2^ = 0.628).

## 4. Discussion

In this study, we sequenced the genomes of 101 sheep with high and low levels of milk production. Using a neighbor-joining-tree analysis, we divided these sheep into 10 groups according to their genetic relationships and background, and then, using a principal component analysis and a structure analysis, these 10 groups were divided into three clusters: high-milk-yield sheep (EFR, DM, DMF1, DMF2), Mongolian low-milk-yield sheep (HS, CS, AWS, STHS), and low-milk-yield sheep (SFK and FS). The scanning of the genomes of high- and low-yield sheep breeds revealed that the *FCGR3A ARNT*, *CTSK*, *CTSS*, and *GHR* genes were the strongest candidates. According to the RT-qPCR experiment and the analysis of the relationship between these candidate genes’ ΔCT values and milk production, the *FCGR3A* gene had a significant relation with Lacune sheep milk production. All of these genes have also been reported to be highly expressed during the lactation period in cattle [42] and buffalo, reflecting the similar biological functions of these genes when they are expressed in lactating mammary glands. The GO and KEGG analyses showed that most of these genes were significantly enriched for mammary-gland-specific GO terms (cytokine-receptor activity and protein binding), as well as establishing the cellular functions and protease bindings. These genes were found to have high and low levels of milk production, milk protein, milk fat, milk lactose, cheese traits, somatic cell counts, etc. For some of these traits, it is not easy to find a direct relationship with milk quantity. This was probably the reason why we did not find a significant correlation coefficient between the *CTSK*, *CTSS*, or *ARNT* expression values and milk production. Furthermore, as we found in many publications, some of the most significant genes in our research play a crucial role in sheep-mastitis traits and in the immune-response process, which improves milk production. Sheep-mammary-gland mastitis is also a common source of economic losses on sheep farms, so it is also important for us to focus on some of the genes related to mammary-disease resistance.

### Candidate Genes Potentially Associated with Some Milk Traits

The *FCGR3A* (Fc fragment of IgG, low-affinity IIIa, receptor) gene is considered a novel and promising candidate for relieving stress, inflammation, and disease [43], as well as dairy-cattle mastitis. The binding of FCGRs to the Fc region of immunoglobulins mediates a variety of immune functions, such as antigen presentation, the clearance of immune complexes, the phagocytosis of pathogens, and cytokine production [44], they work together to resist viral injection to improve udder health and, thus, increase the production of milk. It is reasonable to suggest that lactation periods cause the increase in expression of a large number of genes, resulting in improvements in performance, as mastitis or Staphylococcus aureus infections often occur during lactation [45,46]. With regard to *CTSS* (Cathepsin S)-encoding protease, Sodhi et al. (2021) found a higher expression of most of the cathepsin genes (*CTSS*, *CTSD*, and *CTSK*) during the mid-to-late lactation stages, emphasizing their potential roles in milk synthesis, since the expressions of most of the proteases were higher during peak lactation [47]. The higher expression of cathepsin-encoding genes during late lactation stages could be attributed to the fact that *CTSS* plays a crucial role in mammary-gland involution [48]. Previous studies found the role of the *CTSK* gene in influencing cheese traits, as it is a protease-encoding casein that increases milk protein [49]. Similar results were found when evaluating the expression patterns of important protease-pathway-associated *CTSK* genes derived from different lactation stages in Sahiwal cows and Murrah buffalo. In both breeds, the RNA-expression levels of these genes were higher in the late lactation stages than in the early lactation stages. Lactation induces bone loss in order to provide sufficient calcium in milk, and to prevent this from taking place, the *CTSK* gene elevates its expression to increase osteocyte numbers, in order to maintain the balance of bone and milk calcium [50,51]. This proves that the *CTSK* gene plays an important role in milk-calcium traits. Furthermore, a differential expression analysis of Churra and Assaf sheep allowed us to notice some genes that were significantly and differentially expressed between the two breeds. These genes were mainly associated with protein-protease activity. Furthermore *CTSK* was differentially expressed and was selected as a candidate gene associated with cheese traits in this study [49]. These findings confirmed previous results, which highlighted the importance of the expression of genes encoding for some proteases in sheep milk. The aryl hydrocarbon receptor nuclear translocator (*ARNT*) can interact with the AHR aryl hydrocarbon receptor). The AHR is restricted to the cytoplasm in its unbound state [52,53]. Once activated, the AHR is combined with the nucleus of the *ARNT* and forms an active complex with the AHR nuclear translocator (*ARNT*) to alter the expressions of target genes [54]. It was reported that an association between AHR activation and *ARNT* causes changes in milk production [55]. More specifically, pregnant mice exposed to *TCDD* (an *ARNT* agonist) in vivo produced lower levels of the milk proteins, β-casein and whey protein [56]. The *GHR* is a regulator of developing growth and, as a growth hormone, it has important effects on carbohydrate, protein, and lipid metabolism. In cattle, mutations in *GHR* have been associated with milk yield and composition in Ayrshire, Holstein, and Jersey cattle. Dettori et al. (2018) reported that variations of the ovine *GHR* gene might affect milk-quality traits in Sarda sheep [57]. The rs55631463 *GHR* SNP genotype affects milk fat and protein yield, and the rs411154235 SNP is associated with lactose content in milk, as it encourages the transformation of glucose into glycogen to help glycogen cellular deposits [58].

In our study, except for *FCGR3A*, the strongest gene candidates, *CTSK*, *CTSS*, *ARNT*, were not found to be significantly relevant to milk-production quantity. This was potentially due to the small number of Lacune sheep selected to validate their function. Since our objective was to detect milk-yield genes, not milk-composition genes, it was difficult to directly find the relationships between these genes, which may participate in milk-protein or milk-fat synthesis. Thus, further measures need to be applied in the future to prove that these genes are correlated with other sheep-milk traits. In particular, Dairy Meade should be tested, since this breed, or crosses in which it is involved, comprised a high proportion of the high-milk-yield group.

## 5. Conclusions

In conclusion, our research is the first attempt to report a genome-signature-selection scan revealing genes that are associated with high- and low-milk-yield sheep breeds by using whole genome sequencing. Some of the most significant candidate genes associated with milk yield were identified through a combination of genome-signature-selection scanning and RT-qPCR experimentation. In the high- and low-yield groups, 553 genes were detected that were enriched with significant protein-receptors-combing pathways. Under selective pressure, we selected the four genes, *FCGR3A*, *CTSS*, *CTSK*, *ARNT*, that were most likely to be related to milk-production traits, in order to perform a correlations analysis, in which *FCGR3A* was found to have a significant relationship with daily milk yield, as it participates in the process of immune reaction. Furthermore, since higher gene-expression values mean severer mastitis, they are negatively correlated with milk production. These results were supported by the correlation between the FCGR3A ΔCT values and the estimated values of the corresponding ewes. Therefore, these results could provide a better genetic perspective on the phenotypic differences between different-milk-yield groups for similar studies. Our findings provide an insight into the dynamic characterization of sheep-mammary-gland gene expression, and the identified candidate genes can provide valuable information for future functional characterization, as well as contributing to a better understanding of the genetic mechanisms underlying the milk-production traits in sheep.

## Figures and Tables

**Figure 1 animals-13-01654-f001:**
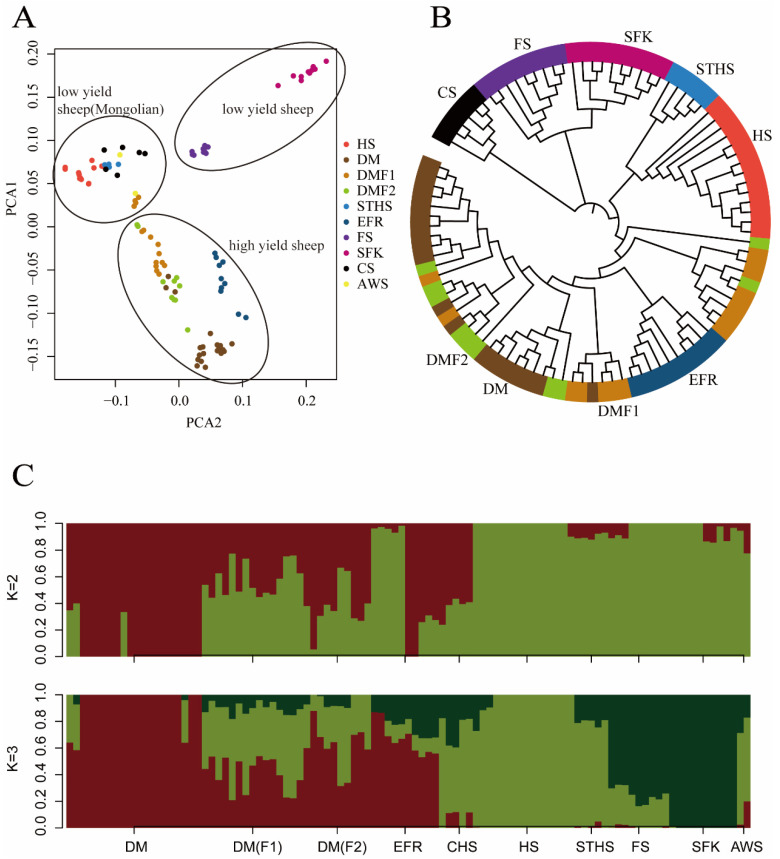
(**A**) PCA analysis of ten sheep breeds, which were divided into three clusters: Cluster1, high-yield breeds (East Friesian sheep (EFR), Dairy Meade sheep (DM), Dairy Meade (DM)F1, Dairy Meade (DM)F2), Cluster2, Mongolian low-yield breeds (Hu sheep (HS), Small-Tail Han Sheep (STHS), Churra sheep (CS) and Awassi sheep (AWS)), and Cluster3, low-yield breeds (Finland sheep and Suffolk sheep). (**B**) The neighbor-joining tree divided one hundred and one sheep into nine different groups. The nine main branches with closest relationships, DM, DMF1, DMF2, EFR, CS, FS, HS, STHS, and SFK, breeds, were clustered into closer bunches. (**C**) Ancestor analysis was performed by ADMIXTURE; when K = 2, all sheep were divided into two groups (high- and low-yield sheep), and when K = 3, all sheep were separated into three groups (high-yield, low-yield, and Mongolian low-yield sheep).

**Figure 2 animals-13-01654-f002:**
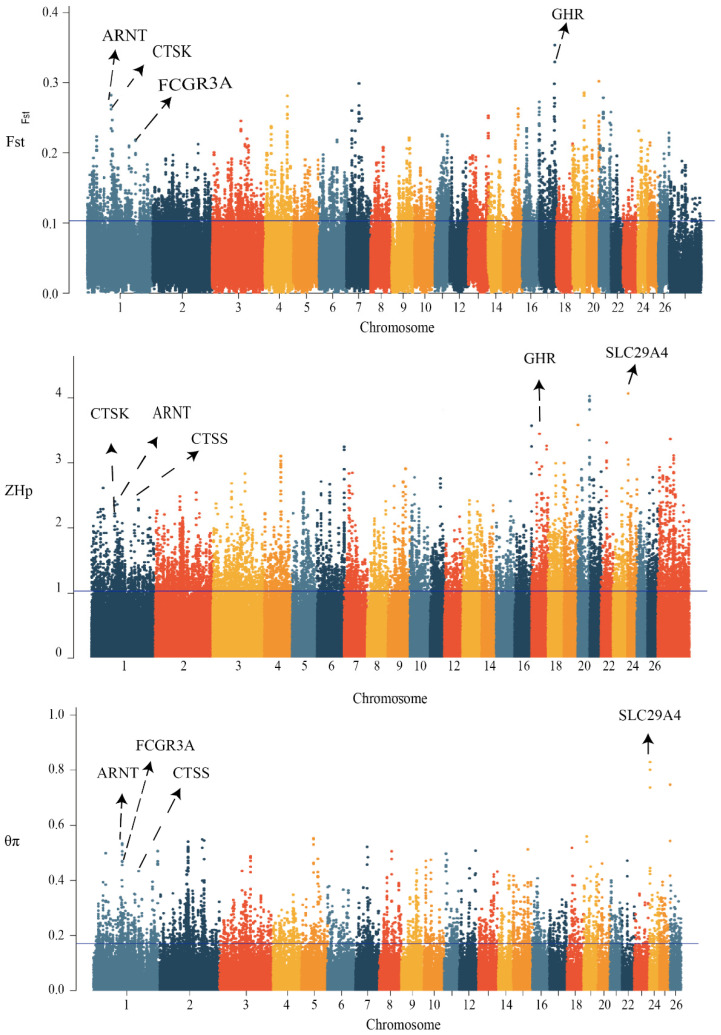
Manhattan plots of genome-signature-selection analysis of total 101 sheep. Three statistical methods were applied to obtain positive selection scans for strong milk-production traits. High-yield sheep (HY) were compared with low-yield (LY) controls. The population genetic differentiation for F_ST_ values, the transformed heterozygosity score ZH_P_, and the nucleotide-diversity θπ ratios (θπ-HY/θπ-LY) were calculated within 100-kb sliding windows. The significance threshold of selection signature was set to top 5% percentile outliers for each individual test and is indicated with horizontal dashed lines.

**Figure 3 animals-13-01654-f003:**
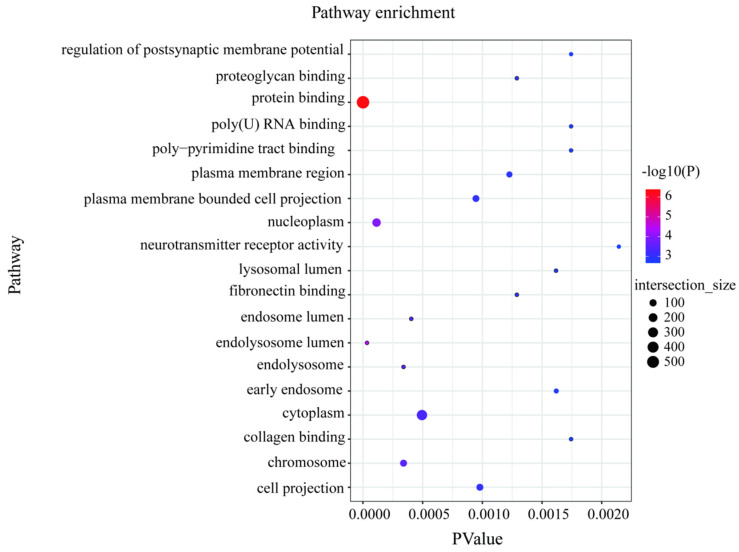
Positive selection of total overlapping genes was performed using functional analysis of Gene Ontology and Kyoto Encyclopedia of Genes and Genomes pathways. The dots colored blue to red means more genes participated in the functional pathway.

**Figure 4 animals-13-01654-f004:**
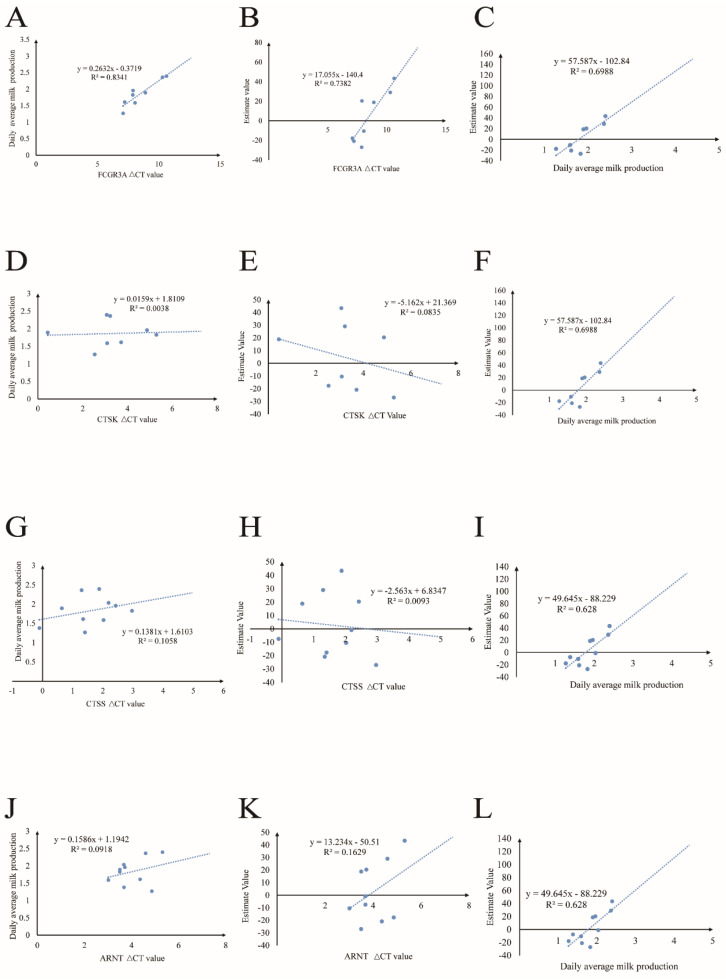
(**A**–**C**) The correlations between FCGR3A gene’s relative expression value and the daily average milk production and estimated values, as well as the relevant coefficient of daily average milk production and estimated values. (**D**–**F**) The correlations between CTSK gene’s relative expression value and the daily average milk production and estimated values, as well as the relevant coefficient of daily average milk production and estimate values. (**G**–**I**) The correlations between CTSS gene’s relative expression value and the daily average milk production and estimated values, as well as the relevant coefficient of daily average milk production and estimate value. (**J**–**L**) The correlations between ARNT gene’s relative expression value and the daily average milk production and estimated values, as well as the relevant coefficient of daily average milk production and estimated values.

**Table 1 animals-13-01654-t001:** 101 sheep information.

Breed	Abbreviation	Number	Milk Yield per Lactation/kg	Data Source	Classification	Reference
Dairy Meade sheep	DM	11	350–600	M-Natural farm	High yield	[18,19]
Dairy Meade (F1) sheep	DMF1	15	350	M-Natural farm	High yield	[18,20]
Dairy Meade (F2) sheep	DMF2	10	500	M-Natural farm	High yield	[18,20]
East Friesian sheep	EFR	10	500–700	NCBI/ PRJNA624020	High yield	[21,22]
Dairy Meade sheep	DM	9	500	NCBI/ PRJNA624020	High yield	[19]
Awassi sheep	AWS	2	300–500	NCBI/ MW260509	High yield	[22]
Small-tailed Han sheep	STHS	5	100	M-Natural farm	Low yield	[23]
Churra sheep	CS	6	100–200	NCBI/ PRJNA395499	Low yield	[24]
Hu sheep	HS	14	100–240	NCBI/ PRJNA624020	Low yield	[25,26]
Suffolk sheep	SFK	10	–	NCBI/ PRJNA624020	Low yield	[27]
Finland sheep	FS	9	–	NCBI/ PRJNA624020	Low yield	[28]

**Table 2 animals-13-01654-t002:** Premier sequences of target genes and reference genes.

Gene	Primer Sequence	Length	Synthesis Scale	Format
FCGR3A-F *	CTTAGGACAAATGAAGGCTCTGA	23	10 nmol	TE-100 um
FCGR3A-R *	CTGCCTCTCCACCACGAAT	19	10 nmol	TE-100 um
ARNT-F	CAGGCCGGGTGGTATATGTC	20	10 nmol	TE-100 um
ARNT-R	TGGAAAGCTGCTCACGAAGT	20	10 nmol	TE-100 um
CTSS-F	AAGCTGGTGTCTCTGAGTGC	20	10 nmol	TE-100 um
CTSS-R	CGCCATTGCAGCCCTTATTC	20	10 nmol	TE-100 um
CTSK-F	ATGCAAGCCTGACCTCCTTC	20	10 nmol	TE-100 um
CTSK-R	CCAGTTTCTCCCCAGCTGT	21	10 nmol	TE-100 um
GAPDH-F	TCGGAGTGAACGGATTTGGC	20	10 nmol	TE-100 um
GAPDH-R	CCGTTCTCTGCCTTGACTGT	20	10 nmol	TE-100 um

*-F means the forward premiers of genes; -R means the reverse premiers of genes.

## Data Availability

The original contributions presented in the study are included in the article’s additional files; further inquiries can be directed to the corresponding author. The raw sequence reads of sheep were deposited in NCBI’s Sequence Read Archive (SRA) database and accessible, through accession no. PRJNA914094, for genomic sequences of 41 dairy sheep.

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
