# Peer review of "Genome-Wide Signal Selection Analysis Revealing Genes Potentially Related to Sheep-Milk-Production Traits"

_animals, 2023, doi:10.3390/ani13101654_

Round 1
Reviewer 1 Report (Previous Reviewer 2)
I think it's very interesting what they found
Author Response
Thank you for your interest and we will do research better in the future.
Reviewer 2 Report (Previous Reviewer 1)
The manuscript is substantially improved. It only needs a style revision according to the journal guidelines. I saw that the authors followed my suggestions. However, there are just some little things to fix.
Line 50: Please write Dairy in lowercase.
Line 72: Please remove "in recent years".
Author Response
Line 50: Please write Dairy in lowercase.
Response: Thank you for your suggestion, i have write "Dairy" in lowercase, it is changed to "dairy".
Line 72: Please remove "in recent years".
Response: Thank you for your advice, i have deleted "in recent years" in the text.
Reviewer 3 Report (New Reviewer)
The MS compared the genetic background of high and low milk-yield sheep groups. Some genes around of the top hits of the comparisons have been selected to test by real time PCR.
Notices:
Line 29: '...We chose the highly selected genes...' Please find something better than 'highly selected'.
Line 90: Since the MS has more than one author, I think the proper for is 'In our research,...'
Line 121-122: Instead of '...after quality control, qualified data was then used for...'
please use
'...after quality control, data was used for...'
Line 123: '...from NCBI were analyzed by Plink on Linux.'
It would be nice to read information about PLINK parameters and the version of PLINK. Or is it described in Section 2.3? If so, please refer that section.
Line 135: '...(5) the linkage disequilibrium locus are excluded'
What was the level of the exclusion criteria? What was the measurement process of linkage disequilibrium?
Line 177: '...randomly select 11 Lacaune sheep in Belgium farm...'
Dairy Meade would also be interesting to see in that experiment.
Line 214: '...BLUPf90+, in my research, the animal...'
Properly; '...BLUPf90+, in our research, the animal...'
Line 227: '...were able to explain most of the variation.'
How much is the 'most'? Percentage? Eigenvalues of the PCs?
Line 237: '...geographical origin and blood communications of these...'
'Blood communication' is a strange phrase to me. Are the authors sure to want to use that phrase? What does that mean?
Line 312: '...every gene ex-312 pression value was got from 11 ewes, the...'
Comparing the effort put into the NGS and the effort put into RT-qPCR, here I would wait for more.
For example: more animals, and more Dairy Meade animals and not just 11 Lacaune ewes. Hopefully that is the next step to be published in an upcoming article.
Line 418: 'In my study, except...'
Properly: 'In our study, except...'
Line 424: '...further measures need to be done to prove these genes have correlations with other sheep milk traits in the future.'
Please add something like this; especially Dairy Meade is to be tested, since this breed or its cross composed a high portion (36 animals) of high milk-yield group (57 animals)
Author Response
Please see the attachment.

This manuscript is a resubmission of an earlier submission. The following is a list of the peer review reports and author responses from that submission.
Round 1
Reviewer 1 Report
Comments to the authors are available in the pdf file.

Reviewer 2 Report
I would like to know why they chose so many breeds?
Why few samples of some breeds?
Why they included breeds without milk yield?
Wouldn´t it have been better to have more samples of fewer breeds?
Reviewer 3 Report
General comments
The study presents a Genome-Wide Signal Selection Analysis to reveal genes that are potentially related to sheep production traits. There is a novelty behind this idea and the results could potentially offer important information for candidate genes to improve milk yield and quality traits.
However, the study-design is not clear and makes it difficult for the reader to follow, while there are significant inconsistencies and gaps rendering the understanding of the work challenging. More significantly, the numbers of the high- and low-yielding animals studied from the M-natural farm were not sufficiently balanced and only five animals from the low-yielding group were involved. The rest of the sheep genomes (n=60) were downloaded from the NCBI data base. A major drawback here also is the fact that the specific selection criteria of both the 41 studied animals and the 60 acquired genomes are not adequately justified (for example Suffolk, a meat breed of sheep is also included in the study, whereas, only 2 genomes from the Awassi breed were used)
A moderate english editing is required across the manuscript
Other comments:
The objectives of the study should be more clearly stated and added at the end of the last paragraph in the Introduction section.
In several parts of the text formatting issues exist (e.g., missing gaps between words, missing full stops and commas)
Authors should avoid long sentences and split them into shorter ones; in many parts it is really difficult to follow the text.
When presenting the results past tense should be used rather than present tense (e.g., lines 31-33)
Specific comments
Abstract
Lines 33-35. A stronger and more specific statement for closing the abstract is needed.
Introduction
The Introduction is poorly written and several phrases from other publications have been used with just minor paraphrasing.
The same occurs for the description of specific parts in the Materials and Methods section (e.g., many sentences seem to be the same with text from the publication by Liu X, Zhang Y, Liu W, Li Y, Pan J, Pu Y, Han J, Orlando L, Ma Y, Jiang L. A single-nucleotide mutation within the TBX3 enhancer increased body size in Chinese horses. Curr Biol. 2022 Jan 24;32(2):480-487.e6. doi: 10.1016/j.cub.2021.11.052. Epub 2021 Dec 13. PMID: 34906355; PMCID: PMC8796118)
Line 42. Delete ‘in the Mediterranean area’.
Line 44. Add a space after ‘Europe’
Lines 46-50: Too long sentence. Split it an rephrase to be clearer.
Lines 50-52: The reference here has been published 10 years ago. Please provide an updated reference to describe the current status.
Lines 54-55: Delete ‘in these countries’
Lines 55-57: For the intensive systems in these countries the average milk production is much higher than the stated 85-216 Lt/ewe/lactation. A better revision of the literature on this aspect is needed.
Lines 58-60: Rephrase ‘Besides, for the nutrition value of sheep milk, it has higher fat, protein, lactose, ash and total non- fat solids content and has higher nutritional value comparing to cow and goat(Jandal 1996), which could be processed into various delicious milk products.’ to ‘Sheep milk, has higher fat, protein, lactose, ash and total non-fat solids content and a higher nutritional value comparing to cow and goat milk (Jandal 1996), which makes it suitable for processing into various types of dairy products.’
Lines 63-66: Better writing is needed here.
Lines 66-75: That is irrelevant with the study.
Lines 84-85: Check the units used to express milk yield. They cannot be in ‘kg/d’; also, give information about the milk yield of the other breeds as well.
Material and methods
The authors say that 101 ear tissues were sampled, implying that this number of animals were enrolled, but in the next sentence they describe only 41 ewes that were actually sampled in the study. Data from 60 more ewes from other breeds were downloaded from NCBI. The authors need to better clarify how many animals were actually sampled.
Numbers of high- and low-yielding animals are not appropriately balanced (36 vs 5). Moreover, data from both purebred and crossbred animals are considered together with data from only purebred animals acquired from the NCBI data base. In general, the authors need to better clarify how they selected the animals involved in the study, and the animals selected from the NCBI data base. For example, why did they consider only 2 animals of the Awassi breed? (this could also explain the fact that Awassi ewes were assigned to the second cluster ‘Mongolian low yield breeds’)
In Table 1 the animals sum at 102 rather than 101.
Line 110: Suffolk breed is a meat sheep breed and not a low milk-yield dairy sheep breed
Discussion
A more comprehensive discussion of the results and a better literature review of updated literature is needed.